# Current and Future Treatment of Mucopolysaccharidosis (MPS) Type II: Is Brain-Targeted Stem Cell Gene Therapy the Solution for This Devastating Disorder?

**DOI:** 10.3390/ijms23094854

**Published:** 2022-04-27

**Authors:** Claire Horgan, Simon A. Jones, Brian W. Bigger, Robert Wynn

**Affiliations:** 1Blood and Marrow Transplant Unit, Department of Paediatric Haematology, Royal Manchester Children’s Hospital, Manchester University NHS Foundation Trust, Manchester M13 9WL, UK; claire.horgan@mft.nhs.uk; 2Willink Unit, Manchester Centre for Genomic Medicine, Manchester University NHS Foundation Trust, Manchester M13 9WL, UK; simon.jones@mft.nhs.uk; 3Stem Cell and Neuropathies, Division of Cell Matrix Biology & Regenerative Medicine, University of Manchester, Manchester M13 9PT, UK; brian.bigger@manchester.ac.uk

**Keywords:** mucopolysaccharidosis type II, bone marrow transplant, stem cell gene therapy

## Abstract

Mucopolysaccharidosis type II (Hunter Syndrome) is a rare, x-linked recessive, progressive, multi-system, lysosomal storage disease caused by the deficiency of iduronate-2-sulfatase (IDS), which leads to the pathological storage of glycosaminoglycans in nearly all cell types, tissues and organs. The condition is clinically heterogeneous, and most patients present with a progressive, multi-system disease in their early years. This article outlines the pathology of the disorder and current treatment strategies, including a detailed review of haematopoietic stem cell transplant outcomes for MPSII. We then discuss haematopoietic stem cell gene therapy and how this can be employed for treatment of the disorder. We consider how preclinical innovations, including novel brain-targeted techniques, can be incorporated into stem cell gene therapy approaches to mitigate the neuropathological consequences of the condition.

## 1. Introduction

Mucopolysaccharidosis type II (Hunter Syndrome) is a rare, x-linked recessive, progressive, multi-system, lysosomal storage disease caused by deficiency of iduronate-2-sulfatase (IDS) [1]. Under normal circumstances, IDS catalyses the removal of the sulphate group at the 2 position of L-iduronic acid in dermatan sulphate and heparan sulphate [2]. In its absence, these glycosaminoglycans (GAGs) accumulate within lysosomes, leading to progressive cellular dysfunction. The estimated incidence of MPSII is around 1.3 per 100,000 live male births [3,4]. Affected individuals are almost always male, although a few female cases with the disorder have been reported as a result of chromosomal rearrangements [5,6].

MPSII is clinically heterogeneous and has traditionally been described as either a severe or attenuated form based on the length of survival and presence, or absence, of neurological disease. It is becoming increasingly clear that the disease exists as a continuum between the two forms, with disease severity linked to relative levels of IDS enzyme [7,8].

The condition arises because of mutations in the IDS (iduronate 2-sulphatase) gene, which maps at the chromosomal region Xq28 [9]. This gene encodes for a 550 amino acid polypeptide that is processed into a mature protein, the IDS enzyme. The IDS enzyme catalyses hydrolysis of the C2-sulphate ester bond of 2-O-sulfo-α-L-iduronic acid residues in dermatan and heparan sulphate [2].

The clinical phenotype of MPSII is a consequence of progressive pathological storage of GAGs in nearly all cell types, tissues and organs. The majority of patients present with a progressive, multi-system disease, which is generally diagnosed between the age of 18 and 36 months, or earlier in those individuals with an affected sibling. Patients with an attenuated form tend to present between the ages of 4 and 8 years, with considerable overlap [7]. 

Affected individuals usually appear normal at birth, although they may be large, with inguinal and umbilical hernias, and an increased incidence of Mongolian blue spots [10]. The typical coarse facial features associated with the condition usually appear within the first 3 years. Other presenting features include skeletal abnormalities, hepatosplenomegaly, macroglossia, enlarged tonsils and adenoids, upper airway obstruction, recurrent ear infections and cardiac valve abnormalities [11,12]. Early developmental milestones may be normal, but some patients will fail hearing screens in their first year, and speech delay is not uncommon. In severe cases, developmental delay is usually apparent by 18 to 24 months, with slow progress after this stage and a developmental plateau between the age of 3–5 years. The velocity of regression observed can be more complicated to predict due to the prolonged plateau of developmental stagnation that can last several years.

In contrast to the characteristically placid nature of children severely affected with MPSI, patients with MPSII can be hyperactive and aggressive. Experts have sought to ascertain more about the nature of this behavioural phenotype and have suggested that limited communication skills, frustration, anxiety, sleep disturbance, sensory-seeking behaviour and poor emotional regulation all contribute [13,14]. Following their developmental plateau, patients with central nervous system (CNS) involvement suffer a progressive neurological decline, rendering them severely handicapped and entirely dependent on caregivers by the time of their death [7]. Progressive cardiorespiratory compromise occurs due to GAG deposition in the upper airways, thoracic skeleton, heart, liver and spleen, and loss of neurological function, and it results in death in the second decade [15]. 

In contrast to the relatively predictable clinical course for severely affected patients, individuals with the attenuated form tend to present later and remain cognitively stable into their third or more decade [16]. They are still, however, subject to the same multi-system pathological processes that occur in severe forms and may still have symptoms and complications leading to significant morbidity and disability. These include mild to moderate learning difficulties and broader neurocognitive struggles outside of the educational environment, such as deficits in attention and visual–motor skills. The deterioration in cardiorespiratory function tends to follow a more protracted sequence than in severely affected patients and leads to death between the ages of 20 and 30 years [17]. The term ‘neuronopathic’ is used by some authors and investigators when referring to the severe phenotype of MPSII, which is typically associated with the characteristic neuropathology experienced by affected individuals. For purposes of this review, severe disease will be referred to as severe throughout apart from when referring to articles that have used the term ‘neuronopathic’, in which cases ‘neuronopathic’ will be used.

The rarity of MPSII combined with considerable heterogeneity in disease presentation poses a significant challenge, since patients may present to several different specialists before a unifying diagnosis is made.

## 2. Current Therapies for MPSII

### 2.1. Enzyme Replacement Therapy 

Enzyme replacement therapy (ERT) serves to correct the metabolic defects associated with MPSII through the intravenous administration of a functional recombinant version of the deficient enzyme. The US Food and Drug Administration approval of idursulfase in 2006 and subsequent approval of idursulfase beta by the Korea Food and Drug Administration in 2012 [18] transformed the treatment of the condition and has been proven to reduce urinary GAG levels and liver and spleen volumes in MPSII patients [19]. These improvements are sustained at 8 years post ERT start, and real-world data also suggest that idursulfase therapy improves other somatic cardio-respiratory parameters, including left ventricular mass index and the 6-min walk test distance, and stabilises predicted forced vital capacity and forced expiratory volume in 1 s [20]. A recent retrospective review of ERT-treated MPSII patients found a reduced need for neurosurgical intervention in the first 8 years of life as well as improved survival [8].

The effectiveness of ERT for MPSII patients, however, is significantly limited by its inability to cross the blood–brain barrier (BBB) and thereby address the neurological manifestations of the disease [21]. Since approximately two-thirds of MPSII patients have progressive cognitive impairment [8], there is a great unmet need for suitable treatment options. ERT appears to shift the phenotype of the condition by attenuating the disease course. The huge impact that ERT has on somatic manifestations, including the cardiovascular complications typically associated with the early mortality of affected individuals, results in patients living longer, giving time for cognitive decline to ensue when previously patients would have died from somatic complications prior to this being the case. Cartilaginous organs, including the bones, eyes, trachea, and bronchi are also poorly impacted by ERT, perhaps as a result of poor penetration into the tissues.

Anti-drug antibody formation, occurring in about 50% of MPSII patients treated with idursulfase [22], is another significant limitation of ERT which may reduce its efficacy, particularly in the long term, or lead to hypersensitivity reactions. Other significant limitations of ERT lie in the disease burden it places on families as patients require lifelong intravenous weekly infusions (usually 3 h but may be shortened to 1 h in the absence of infusion reactions). Most patients require permanent vascular access to facilitate this with the subsequent associated infection risk and need for hospital admission with any febrile episode. ERT is generally well-tolerated with an acceptable safety profile, but infusional reactions are not uncommon, affecting around 30% of patients receiving idursulfase [23]. The high cost of ERT is another important consideration and a frequent subject of debate in this era of complicated health economics [24].

### 2.2. Intrathecal or Brain Targeted Enzyme Replacement Therapy

Intrathecal ERT has been proposed as a strategy to mitigate the inability of standard ERT to cross the BBB and address the neurological manifestations of MPSII. Calias et al. in 2012 were able to prove this principal in animal models by demonstrating that lumbar intrathecal administration of IDS to enzyme deficient animals reduced GAG storage in both superficial and deep brain tissues, with concurrent morphological improvements [25]. A phase I/II clinical study of 16 cognitively impaired males with mucopolysaccharidosis II found this approach to be safe in the clinical setting with no serious adverse events reported in the 12 patients who received the intrathecal idursulfase preparation through an intrathecal drug delivery device (IDDD). Although surgical revision or removal of the IDDD was required in six of the 12 patients, mean cerebrospinal fluid GAG concentration was found to have decreased by 90% and 80% in the higher and lower dose cohorts, respectively [26]. 

The recent update to an ongoing phase 2/3 extension study evaluating the long-term safety and efficacy of intrathecal idursulfase in conjunction with intravenous idursulfase in children with neuronopathic MPSII reported data on 47 patients [27]. Patients received monthly intrathecal IDS along with weekly intravenous enzyme, and interim analysis was performed at 2 years after patients had completed at least 12 months of intrathecal therapy or discontinued. Efficacy data were summarised by treatment group as early or late determined by whether patients had received early intrathecal therapy (at least 12 months in a previous phase 2/3 study) or delayed in whom patients had not received any previous intrathecal therapy. Intrathecal IDS was generally well-tolerated, and CSF GAG concentrations decreased over time in both groups; however, the key efficacy endpoint of cognitive function was not statistically significant, and the study is ongoing, so final results are not available. 

One of the main difficulties in interpreting this data could be the fact that it is very difficult to predict what an individual’s MPSII neuropathic phenotype will be, which renders assessing the impact of such therapies on cognitive function very challenging.

Several IDS-fusion solutions for crossing the blood–brain barrier have recently been proposed for ERT and are undergoing testing in clinical trials including JR141, which is a fusion of IDS with the anti-human transferrin receptor antibody, preventing neurodegeneration and neurocognitive dysfunction in MPSII mice [28] and a significant reduction of HS in the CSF, suggesting successful penetration of the blood–brain barrier in a phase I/II trial [29]. This therapy is now approved in Japan. AGT-182, (NCT02262338) an alternative ERT approach using IDS fused to a monoclonal antibody against the human insulin receptor (HIR) protein, and DNL310 (NCT04251026), an IDS transferrin receptor binding domain fusion, are both currently in trial.

### 2.3. Substrate Reduction Therapy

Substrate reduction therapy (SRT) aims to prevent the synthesis of the compounds that pathologically accumulate in the absence of a specific lysosomal enzyme. Most of the current approaches for LSDs use a small molecule drug analogous of intermediate compounds in the biosynthesis pathways which serve as functional competitors that limit the number of molecules requiring catabolism within the lysosome. At present, SRT is approved to treat some LSDs in the US and Europe including Gaucher disease type 1 and Neimann–Pick type C [30]. 

The problem with SRT for mucopolysaccharidoses is that many of the intermediates involved in GAG synthesis are also involved in several other essential metabolic pathways, so analogues could potentially interfere with many other crucial metabolic processes, leading to serious consequences [31]. For this reason, current strategies for MPSII focus on the indirect inhibition of GAG synthesis [1]. In 2006, genistein, a compound that can be purified from soya beans and acts as a tyrosine kinase inhibitor, was found to reduce in vivo GAG storage in fibroblasts of MPS I, II, III, IV and VII cells [2]. Friso et al. in 2010 went on to demonstrate a reduction in urinary GAG levels in mouse MPSII models after 10 weeks of genistein treatment as well as a reduction in tissue samples from the liver, spleen, heart and kidneys. Decreased GAG deposits in the brain after genistein were also found in some animals [3]. Genistein has also been shown to improve connective tissue elasticity and joint range of motion in a small cohort of MPSII patients [4].

The main potential advantage of SRT with genistein is its ability to cross the BBB and therefore, it could be used in a combined therapeutic approach with ERT; however, clinical trial results for a neurological benefit in MPSIII children have so far proved disappointing [1,5]. The first double-blinded placebo-controlled trial designed to specifically assess whether a high dose of genistein improves neurological symptoms in MPSIII patients found that although genistein appears to circulate in significant amounts in the blood after dosing, and be associated with a small but significant reduction in urinary GAGs, there is no evidence of any benefit in neurocognition or psychological well-being of individuals or families [6]. At present, SRT is not approved for any of the MPS disorders.

## 3. Bone Marrow Transplant for MPSII

### 3.1. Introduction

Haematopoietic stem cell transplantation (HSCT) for lysosomal storage disorders uses the principle of cross correction, whereby lysosomal enzymes secreted by healthy cells can be used to cross-correct neighbouring enzyme-deficient cells [7]. HSCT provides the recipient with a continuous source of enzyme from donor-derived myeloid cells and was first successfully described in the clinical setting for a patient with MPSI (Hurler syndrome) in 1981 [8]. Patients must first be conditioned with a full intensity myeloablative regimen to create space in the recipient’s marrow for the infused donor stem cells to engraft, and they must be immune suppressed so that the transplanted cells are not rejected. Full-intensity myeloablative conditioning with fludarabine and pharmacokinetic-adjusted parenteral busulfan is the current recommended conditioning regimen for LSDs [9]. Therapeutic drug monitoring of the administered intravenous busulfan permits more precise dose delivery, thereby mitigating the previously high rates of veno-occlusive disease (VOD) associated with increased busulfan exposure whilst ensuring adequate therapeutic levels are achieved to avoid graft rejection [10]. 

Umbilical cord blood (UCB) is the preferential stem cell source for LSDs, as it is associated with superior levels of full donor chimerism compared to other stem cell sources [11], resulting in higher levels of enzyme delivery and subsequent improvement in disease-related outcomes [12]. Additionally, the better tolerance of HLA mismatch and shorter time to transplant that UCB affords enables patients to be transplanted at a younger age to further optimise their treatment response. Immune-mediated cytopenias and primary graft failure have been associated with UCB transplant in this setting [13] and remain an area of active investigation; however, rates have declined significantly over the past few decades, and recent evidence suggests that B-cell depletion with the addition of rituximab to conditioning regimens offers a promising solution [14].

MPSI is the paradigm of successful HSCT in LSDs and is the gold-standard treatment for patients under the age of 2 years with severe forms of the disease and no, or minimal, cognitive impairment [15]. It confers several advantages over enzyme therapy, including the potential to mitigate the neurological manifestations of the disease through the migration of donor-derived cells across the blood–brain barrier. These cells differentiate into tissue macrophages, known as microglia, which are able to secrete functional enzyme and correct deficiency in the central nervous system [16]. Furthermore, HSCT provides a durable lifelong enzyme source to the recipient, abrogating the need for frequent hospital attendance for lengthy enzyme infusions as well as the financial and quality of life burden of such treatment. The HSCT process replaces the recipient’s immune system with that of the donor so anti-enzyme antibodies are not an issue. 

### 3.2. Summary of Bone Marrow Transplant Outcomes in Hunter Syndrome

Despite the increasing body of evidence for the effectiveness of HSCT in Hurler syndrome, the overall experience for MPSII is limited, and much of the available literature is outdated [17]. A systematic review of the data is hampered by the fact that MPSII patients are often evaluated as part of large heterogeneous cohorts of patients transplanted for various metabolic disorders or limited to case reports and small case series with variable results. 

The first application of HSCT for MPSII in 1986 showed normalisation of IDS activity in leukocytes and stabilisation in the cognitive function of a 7-year-old patient. Plasma IDS activity, however, remained well below the normal range, and the patient died due to cardiovascular complications 3.5 years after transplant [18]. The next available literature on HSCT in MPSII is from three separate case reports in 1994, two of which were on older children, aged 14 years and 9 years 10 months, with non-neuropathic phenotype [19,20]. These patients showed somatic improvements including in hepatomegaly, cutaneous manifestations, joint contractures and cardiac valve dysfunction, with the 14-year-old returning to school 7 months post-procedure and continuing to show gains in intellectual function. The same year, Miniero et al. reported a case of a 31-month-old with neuronopathic MPSII who underwent HSCT. They demonstrated the safe use of G-CSF along with stabilisation and some improvements in somatic functions; however, there was no mention of the cognitive impact of HSCT in this individual compared to the natural history of the disease. 

The cases that followed showed limited, and conflicting, evidence for the utility of HSCT in addressing the neurocognitive issues associated with MPSII. In 1995, Coppa et al. [21] reported the 2-year follow up of a patient who had been transplanted aged 2 years and 9 months old. Despite stabilisation in brain MRI appearances and audiometry compared to the pre-transplant state, the longitudinal neuropsychological evaluation showed significant worsening after the third month post-transplant. There was partial recovery in motor and social skills on assessment twenty months post, but verbal performance remained unchanged. McKinnis et al. [22] also failed to show the neurological benefit of HSCT in a child who had been transplanted aged 29 months. Serial biopsies demonstrated persistent GAG deposition in neural structures in contrast to the reduction in GAG deposition observed in non-neural structures, and the patient suffered from a progressive neurological decline such to the extent that at 8 years old, the patient’s intellectual level was consistent with that of a 10-month-old. In contrast to this, a further case reported by Li et al. [23] around the same time showed normalisation in plasma IDS activity and maintenance of leukocyte IDS levels around 60% of normal values in a patient 4 years post-transplant who had undergone HSCT at the age of 5 years. This coincided with somatic improvement and stabilisation in neurocognitive and cardiac function.

Since this time, several other case series and reports have sought to evaluate the efficacy of HSCT in MPSII with variable results, particularly regarding the neuropathophysiological impact of the condition. Conclusions and limitations of the main literature available are summarised in Table 1 with a more detailed overview of each study given below.

In 1999, Vellodi et al. [24] reported long-term follow-up outcomes of a 10-patient cohort who had been transplanted between 1982 and 1985. Of the 10 patients, only 3 had survived more than 7 years post-transplant, which the authors felt could largely be attributed to poor donor selection. Four of the patients had died within 100 days of transplant from complications, with a further patient dying of bronchiolitis obliterans 4 years post and another death due to GVHD. A seventh patient autologously reconstituted and was subsequently lost to follow up. Of the three patients who survived more than 7 years, two patients continued to show steady progressive physical and intellectual disabilities. One of these had been transplanted aged 20 months from a matched sibling donor, who later turned out to be a carrier of the condition, and the other patient had been transplanted aged 5 years and 1 month from a matched unrelated donor. The third patient, who had been transplanted aged 10 months after being screened because of disease in the family, showed stable neurological function and, despite having some concentration issues and a borderline IQ, was able to attend mainstream school. This was a significantly improved outcome compared to his affected maternal uncle, suggesting that HSCT may have neurological benefit if performed early enough.

This concept was further supported by Maria et al. in 2007 [25], who reported a series of five patients undergoing unrelated umbilical cord blood HSCT aged between 0.26 and 3.4 years with a median follow up of 1.7 to 3.7 years. Four out of the five patients engrafted with full donor chimerism and continued to show gains in cognitive, language, adaptive and motor skills, with the oldest patient having the slowest gains. The fifth patient in the cohort had mixed donor chimerism and died due to GVHD complications post a second HSCT. It must also be acknowledged that there was a huge change with improvements in supportive care for patients undergoing HSCT between the cohorts reported by Vellodi et al. in 1999 and Maria et al. in 2007.

In 2009, Guffon et al. [26] reported long-term follow up data, ranging from 7 to 17 years, on a series of eight boys who had been transplanted between the ages of 6 and 16 years from 1990 to 2000. Six of the patients were transplanted using HLA-matched sibling donors, two of whom were heterozygous for the IDS mutation, and the remaining two patients were transplanted using an HLA-matched and a HLA-mismatched unrelated donor. All patients transplanted from healthy homozygous donors had normalisation of leukocyte IDS activity, with around 50% activity observed in the two patients transplanted from carrier siblings, but IDS activity in the serum remained very low for all patients. This probably reflects that in a healthy individual, IDS is secreted by the liver as well as leucocytes, so that in the transplanted Hunter patient, leukcocyte activity should be normal, but plasma levels would be expected to be below normal. There were improvements reported in many somatic features of MPSII including hepatosplenomegaly, upper airway obstruction, coarse facial features and urinary GAG excretion, and stabilisation in cardiovascular function for the duration of the follow-up period. Neuropsychological outcomes were highly variable. Two patients who were transplanted with an attenuated phenotype reached adulthood with normal IQ, social, school and language development, whereas the four patients who had significant cognitive impairment prior to transplant (IQ/DQ < 80) continued to show progressive neurological deterioration following HSCT. Three patients lost the ability to walk in their early teenage years, two developed epilepsy, two lost language and all four patients required special schooling. 

A study by Poe et al., reported in 2011, followed nine patients undergoing umbilical cord HSCT between the ages of 1.5 months to 3 years and 11 months for MPSII and found some evidence of neurological benefit from HSCT, although it concluded that delays were still apparent [27]. Follow-up ranged from 7 months to 7 years, with five of the seven living patients continuing to show gains in some, or all, of the developmental domains, and one patient had normal development in four of the six domains evaluated. Escolar et al. reported further data in 2012 [28] in which the developmental status of these nine patients were compared with a cohort of 35 patients who had not been transplanted. Patients were assessed up until the age of 8 years. Children who were transplanted at a younger age (<18 months) had better outcomes than those transplanted at an older age and showed normal to near normal development of cognitive, adaptive and language skills. Children transplanted early showed continuous gains in these skills, although at a slower rate than normal children, whereas those transplanted after 18 months reached a plateau before regressing to a functional age of 1 to 3 years. This study suggests that umbilical cord transplant in younger MPSII patients does have the potential to mitigate at least some of the neuropathological effects of the condition, although more consistent data are needed. 

A Japanese retrospective study from Tanaka et al. in 2012 [29] described the follow-up of 21 patients transplanted with MPSII from 1990 to 2003. Age at time of transplant ranged from 2 years to 19 years and 8 months, and mean follow-up period was 9.6 ± 3.5 years. The authors concluded that HSCT was effective on brain involvement in MPSII if performed before the onset of developmental delay and brain atrophy; however, they acknowledged that this might not be the case for most severely affected patients. Despite the study being limited by its retrospective nature, HSCT also seemed to be effective on cardiac involvement if performed before valve regurgitation developed and activities of daily living (ADLs) remained at baseline values. As with other studies, urinary GAG levels were also lower in HSCT recipients than in ERT-treated patients.

The long-term follow-up of four patients post HSCT for neuronopathic MPSII reported by Annibali et al. in 2013 [30] showed a much slower rate of neurological regression than would be expected, providing further evidence in support of the potential cognitive benefit afforded by HSCT if performed at the right time in selected patients. Serial assessments showed stable IQ for 5 years post-transplant in three of the four patients, with the fourth patient remaining stable for 8 years prior to neurological regression. All patients had mild or moderate intellectual disability prior to HSCT (IQs ranging from 49 to 70) and demonstrated improvement or stabilisation in somatic function post.

More data come from a 10-year report published by Wang et al. in 2016 on 34 patients transplanted for mucopolysaccharidosis in China [42]. Of these, 12 patients had MPSII and were transplanted between the age of 2 and 6 years. Patients all received a busulfan-based myeloablative conditioning regimen, and most of the whole cohort (91.2%) achieved full donor chimerism. Follow-up evaluation showed multi-system somatic improvements including in airway obstruction, joint stiffness, hepatomegaly and recurrent otitis media. Considering the neurological impact of HSCT, four of the 12 patients showed significant improvements in motor skills, and two showed some gains in language abilities. The authors concluded that allogeneic HSCT is beneficial for the neurological development of MPSII patients; however, the mean follow-up for the whole cohort was only 2 years, and there is little detail about the cognitive abilities of patients pre-transplant. A significant difference in survival was observed in patients who were transplanted pre and post 2009 (55.6% vs. 95.7%, *p* = 0.02) and although outcome did not appear to correlate with graft source, it must be acknowledged that most cord transplants (54.5%) were carried out before 2009.

A large retrospective study by Kubaski et al. in 2017 [43] compared data from 146 patients transplanted for MPSII, including 27 new cases and 119 published cases, and compared them with 51 ERT and 15 untreated cases. Of these, 74% of patients had severe MPSII with the remaining 11% having an attenuated phenotype. Despite the study being significantly limited by the age at time of transplant, which ranged from 2 to 21.4 years in the previously unpublished patients, the authors concluded that the presented data supported the positive effect of HSCT on neurological outcomes for MPSII patients evidenced by reduced degeneration on brain MRI and more favourable outcomes in cognitive functions. Additionally, a greater number of transplanted patients experienced improvements in somatic symptoms, activities of daily living and joint stiffness compared with patients receiving ERT, and HSCT was associated with a more significant reduction in GAG levels. 

More recent evidence for the potential utility of HSCT on the neurological effects of MPSII comes from a case series from Selvanathan, who described four patients who showed neurocognitive stabilisation following transplant [44]. Patients were transplanted between the ages of 8 months and 3 years and 8 months. All four patients showed somatic improvements with three out of four showing neurocognitive improvement, and the fourth case who had neurological sequelae prior to HSCT stabilised after a period of initial deterioration. The two patients transplanted at the youngest age appeared to have the most benefit and were able to attend mainstream school without assistance. These data highlight that patients should be transplanted at a young age, prior to the onset of CNS involvement, in order to obtain the most benefit. 

Alongside the previously discussed literature, other studies have suggested that HSCT may be associated with better ADL outcomes in comparison to ERT-treated patients if performed early enough [45], and that both HSCT and ERT are effective in restoring growth in MPSII patients [46]. A limited study from 2004 also found improvements in dermatological manifestations of the condition post HSCT [47]. 

Overall, the limited and heterogenous data available suggests mixed outcomes for patients undergoing bone marrow transplant for MPSII, particularly considering its ability to mitigate the neurological consequences of the condition. The literature is fraught with variable quality papers, and inconsistencies in patient’s age and pre-transplant status, with very few reports about those patients receiving transplant at a very early stage. The most informative studies are probably those by Maria, Poe and Escolar who do all suggest that transplant, if performed early enough and prior to the onset of significant neurological deterioration, may attenuate the neuropathological disease course; however, there is no evidence to suggest that these responses are durable into adulthood, particularly in those with a severe phenotype. 

### 3.3. Haematopoietic Stem Cell Gene Therapy

Haematopoietic stem cell gene therapy (HSCGT) involves combining the expansion capability of haematopoietic stem cells, which are capable of replacing the entire blood and immune system of an individual, with the capacity for long-term replacement of a defective gene copy using integrated gene therapy vectors [48]. Autologous mobilised CD34+ peripheral blood stem cells from a patient are sent to a centralised transduction facility where they are transduced using a lentiviral vector before being frozen and returned to the transplant centre. Here, they are thawed and transplanted into a fully conditioned patient, and following engraftment, they then traffic around the body to all organs and compartments. This process can be exploited to treat inherited neurological diseases, since gene-modified monocytes traffic to the brain and engraft as microglial-like cells that deliver protein effectively to brain cells (see Figure 1).

### 3.4. Advantages of Haematopoietic Stem Cell Gene Therapy

HSCGT has several potential advantages over allogeneic HSCT, since safe allogeneic transplant relies on the availability of an HLA-matched donor and, for many individuals, there may not be a suitable well-matched donor. This is particularly relevant for disorders such as MPSII that have a higher prevalence in East Asian countries [49], where donor registries are more limited. Using a mismatched donor increases the risk of HSCT in contrast to the autologous HSCGT approach where all individuals can potentially donate their own cells.

As previously discussed, allogeneic HSCT is associated with several risks including graft versus host disease (GVHD), which is clearly not a concern when the patient’s own cells are used as in HSCGT, and infection. The risk of infection is highest prior to immune reconstitution, and the duration of this period has been shown to be shorter following autologous transplant with gene-modified cells compared with allogeneic HSCT [50].

A further advantage afforded by HSCGT, which is particularly relevant for neurological disorders such as MPSII, is the potential to achieve supra-physiological enzyme levels. Better biochemical correction in metabolic disease is associated with more enzyme secretion and therefore better clinical outcomes [51]. Gene therapy offers the potential to achieve supra-physiological enzyme levels, since more gene copies can be delivered into the haematopoietic stem cell, and transcriptional control of the transgene can be altered with an appropriate promoter, enabling enzyme production by a mature leukocyte where it may previously had been silenced [48]. 

Following on from work in MLD (metachromatic leukodystrophy), MPSI, MPSIIIA and MPSIIIB, where supraphysiological enzyme levels achieved after HSCGT have been shown to correct neurological disease manifestations in mouse models [52,53,54,55,56], further improvements to vector design, to allow trafficking of the delivered enzyme across the blood–brain barrier, have been shown to completely normalise brain pathology and behaviour in MPSII mice [57]. Clinical application of autologous ex vivo haematopoietic stem cell gene therapy technology in the human setting is an evolving field in metabolic disorders. The Milan MLD experience has shown promising results with proven safety and efficacy in a cohort of 33 patients. Most pre-symptomatic treated patients achieved stable motor function, and severe motor-impairment-free survival was significantly longer in patients receiving the gene therapy product. Most patients also showed normal cognitive development at ≤8 years follow up [58,59]. These impressive findings have led to aditasargene autotemcel (ex vivo lentiviral vector haematopoietic stem cell gene therapy for MLD) recently being approved as the first lentiviral haematopoietic stem cell gene therapy for reimbursement by NHS England [60]. 

The delivery of HSCGT has also recently been reported to show extensive metabolic correction in peripheral tissues and the central nervous system in a cohort of patients with Hurler syndrome and MPSIIIA [61,62].

The potential financial benefit of HSCGT is another important consideration, since pharmacological ERT for metabolic disorders is expensive and life-long. HSCGT, although associated with high initial costs, provides a life-long source of enzyme for the patient, abrogating the need for frequent routine enzyme infusions and the associated quality of life burden such therapy places on the patient and family.

### 3.5. Vectors

The first clinical application of HSCGT used gamma retroviral vectors in the primary immune deficiencies, X-linked severe combined immunodeficiencies (X-SCID) and severe adenosine deaminase deficiency; however, the promising early results were overshadowed by the occurrence of genotoxic events in the X-SCID patients [63]. The first-generation vectors contained strong enhancers in their long terminal repeats (LTRs) so that transgene integration near cancer-associated genes resulted in unwanted gene transcription and insertional mutagenesis. This prompted the scientific community to develop vectors that facilitated robust gene correction in stem cells whilst possessing a safer integration profile. 

Lentiviral vectors (LV) derived from the human immunodeficiency virus (HIV) were found to have a superior safety and efficacy profile compared to retroviral vectors (RV). From a safety perspective, they are based on a self-inactivating (SIN) configuration to eliminate the LTR promoters, thus significantly reducing the chance of oncogene activation and making them less likely to integrate near the start sites of actively transcribed genes [64]. They can also be adapted to incorporate the use of a physiological gene promoter, thereby augmenting the clinical benefit of the product and further improving safety by reducing the likelihood of downstream gene activation.

Wakabayashi et al. in 2015 showed that ex vivo HSCGT using second-generation lentiviral vectors improved the biochemical abnormalities in MPSII mice in affected tissues including the cerebrum; however, the results showed only a small increase in cerebral enzyme activity of 2.9% compared to wild type [65]. This may be due to only a few transplanted cells being recruited to the CNS, or more likely, insufficient enzyme production in transduced cells.

### 3.6. Brain-Targeted Stem Cell Gene Therapy

Apolipoprotein E (ApoE) targets low-density lipoprotein (LDL) receptors, LDLR-related protein 1 (LRP-1) and scavenger receptor B1, which are highly expressed at the BBB and, through active endocytosis, facilitates the passage of ApoE-modified nanoparticles into brain capillary cells [32]. In 2013, Wang et al. provided proof of concept that fusion of a receptor-binding peptide from ApoE with a potentially therapeutic protein could bind to LDL receptors on the BBB and be transcytosed into the CNS [33]. Bockenhoff subsequently demonstrated increased brain delivery of the lysosomal enzyme arsulfatase A in mouse models of MLD through fusion of the enzyme to ApoE peptide, and that this approach was superior to other brain-targeted peptides in enhancing CNS enzyme delivery [34].

In order to optimise the effectiveness of HSCGT to treat whole body disease in MPSII, Gleitz et al. exploited the potential of the ApoE peptide to mediate transport across the BBB and developed a novel brain-targeted lentiviral vector by combining the myeloid-specific HSCGT approach, that had previously been shown to deliver superior levels of enzyme to the CNS in MPSIIIA and MPSIIIB [55,56], with techniques that improve the ability of somatic IDS enzyme to cross the BBB [57]. This was achieved by using a lentiviral IDS incorporating a myeloid-specific promoter fused to apolipoprotein E (ApoE) with the aim of increasing the efficacy and uptake of IDS (see Figure 2). This brain-targeted stem cell gene therapy mediated the complete normalisation of brain pathology and behaviour in MPSII mouse models, providing significantly enhanced correction compared to IDS as well as correcting HS storage, peripheral inflammation and other somatic disease markers associated with MPSII. 

In addition to the risks of insertional mutagenesis and RCLs previously discussed, there are several other limitations of HSCGT that must be considered. HSCGT requires a patient to be conditioned with myeloablative chemotherapy so that the gene-modified cells can successfully engraft. This is usually achieved with myeloablative busulfan, which has the advantage of being almost purely myeloablative with very little immune suppression. Conditioning protocols incorporating busulfan have resulted in successful engraftment, rapid neutrophil and platelet recovery and few treatment-related complications [35]; however, there is always some risk associated with any such therapy. Pharmacokinetic monitoring and dose adjustment aims to reduce this as much as possible, but organ toxicity, dose-related bone marrow failure, engraftment issues and infection can still occur. 

Gene therapy approaches in metabolic disorders will always be dependent on the ability of transduced cells to traffic to specific sites and produce sufficient enzyme to ameliorate the pathological effects of the condition. From the current literature, although the biochemical results of HSCGT appear promising, whether this translates into addressing the long-term pathology of progressive neurological disorders is yet to be proven. Most studies to date have been on very young children, prior to the onset of significant neuropathology, and there is little evidence that current approaches will be sufficient to reverse pre-existing problems. This issue could be resolved with the widespread introduction of newborn screening, and the availability of curative therapies could further support the argument in favour of this practice [36].

Despite many clinical HSCGT trials providing robust evidence of multilineage engraftment and safe, stable transgene expression, it remains to be seen whether long-term safety and truly permanent disease correction is achieved. Further barriers to evolution of the practice include the high costs of vector manufacturing and cell transduction as well as complex regulatory requirements. 

### 3.7. Adeno-Associated Virus Gene Therapy 

Adeno-associated viruses (AAVs) are small, replication-defective viruses that require assistance from another virus, such as a herpes virus or adenovirus, to replicate [37]. The main limitations of AAV vector approaches lie in their immunogenicity and their risk of generating neutralising antibodies. Their delayed expression in transduced cells is another significant drawback [38], which is particularly relevant for neuronopathic disorders such as MPSII, where preventing irreversible neuropathology through rapid transgene expression and the production of functional enzyme is paramount.

AAV-vector approaches in MPSII have yielded promising results in animal studies [63,64,65]; however, attempts at translating favourable preclinical outcomes to the clinical setting for MPSIIIA and MPSIIIB have so far proved disappointing [66,67]. 

Despite this, studies evaluating the efficacy of direct AAV gene therapy into the CNS for MPSII are underway, with two multicentre phase I/II clinical trials using a recombinant AAV serotype 9 capsid containing the human IDS expression cassette, RGX-121 (NCT03566043 and NCT04571970), currently recruiting. Another approach was to administer three AAV vectors encoding a pair of zinc finger nucleases (ZFN) designed to target the albumin locus in hepatocytes along with a human IDS donor vector [68]. A phase I/II trial SB-913 (NCT03041324) proved transient with little increase in IDS or reductions in GAGs in five out of six patients. AAV delivery and gene editing approaches for MPSII probably require higher dose delivery with immunomodulation for success.

## 4. Conclusions

As the therapeutic landscape for the management of MPSII has evolved over the past few decades, there remains a significant unmet need when it comes to addressing the devastating progressive neuropathological consequences of the condition, and novel strategies must adapt if they are to successfully modify the disease course and transform the lives of affected individuals. 

The limited and heterogeneous data available evaluating the role of HSCT in MPSII suggest that transplant can be an effective therapeutic strategy, in the absence of any other therapy that impacts the brain, if performed early enough, particularly with modern transplant techniques and advances in donor selection. The reluctance of many to offer transplant to MPSII patients may reflect the problematic early experience resulting in transplant generally being discouraged in many Western countries, particularly since the introduction of ERT. The majority of reports discussing HSCT in MPSII are outdated and hampered by a number of common themes including the age of patients at time of transplant, poor donor selection and the onset of neurological regression prior. In addition to this, most of the transplants were carried out several years ago without modern protocols, rendering the practice much higher risk and fraught with significant morbidity and mortality.

The more recent data available, alongside the increasing body of transplant experience in MPSI, suggest that HSCT when performed at a young age, ideally before 2 years and prior to the onset of neurological regression, can be an appropriate treatment option for Hunter syndrome and is currently the only therapy available with the proven potential to mitigate the neuropathological consequences of the condition. To ensure most benefit from the procedure, umbilical cord blood should be the donor source of choice and pharmacokinetic-guided myeloablative conditioning, incorporating busulfan, used to provide optimal enzyme delivery and subsequent disease-related outcomes whilst minimising the risk of graft failure and VOD. 

There are several limitations of HSCT that should be considered when determining if the treatment should be offered to an individual patient. Despite advances in transplant techniques over recent decades, the procedure still carries a significant amount of risk and is associated with considerable morbidity and mortality. This is particularly the case for MPSII patients who may already have CNS involvement at the time of diagnosis in whom there is no current evidence to suggest that transplant will be of benefit. For this reason, patients need to be diagnosed at a young age, which can be very challenging because of the varied presentation and relies on a high index of suspicion or newborn screening, which was recently approved in the US [69]. In MPSII, somatic features are frequently more subtle than in MPSI, so patients are often diagnosed at an older age, contributing to the inferior transplant outcomes. 

The heterogenous phenotype of MPSII is another significant barrier when assessing the role of HSCT in these patients, as it is very difficult to predict for an individual what they would be like without a transplant. This is perhaps not the case for patients who have affected family members if we assume that familial phenotypes are consistent and compare transplanted and un-transplanted family members at the same age. Despite this, it is clear from the literature that even with improved outcomes for HSCT in MPSII when performed at a young age, it is still a condition associated with considerable pathology, underlining the need for more robust HSCT outcome data and novel experimental approaches to optimise outcomes for these patients. Transplant will always be limited by the time taken for engrafted cells to differentiate into microglia and secrete functional enzyme to correct deficiency in the CNS. This is a particular problem for the progressive neuronopathic MPS disorders, as life-limiting neurological sequelae may develop during this period, and transplant efficacy will always be influenced by the enzyme levels achieved in different compartments. 

The limitations of currently available therapy for MPSII, as well as potential issues with new management approaches, are highlighted in Table 2 alongside potential solutions.

Currently available therapies for MPSII are supportive, serving to alleviate symptoms rather than address the pathogenic mechanisms of the disease. HSCGT has the potential to deliver durable, life-long clinical benefits and the advent of an innovative brain-targeted HSCGT vector for MPSII provides further promise by tackling some of the recognised issues with current experience. The recent outcomes of HSCGT in MLD have offered a new hope for families affected with these debilitating monogenic neurological disorders, and the novel brain-targeted approach for MPSII offers further potential to abrogate the progressive decline in neurocognitive abilities, thereby enabling a wider patient cohort to benefit. 

## Figures and Tables

**Figure 1 ijms-23-04854-f001:**
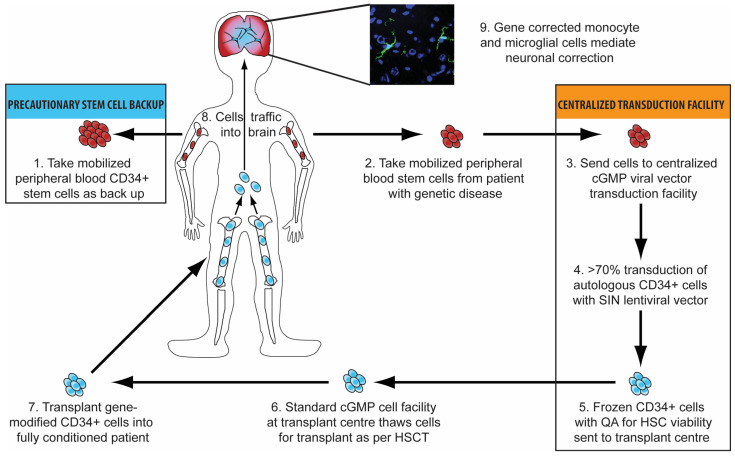
Schematic proposing how autologous stem cell gene therapy can be used to treat inherited neurological disorders. Mobilised peripheral blood CD34+ stem cells are harvested and sent to a centralised transduction facility. Here, they are transduced with an SIN–lentiviral vector before being frozen and returned to the transplant centre. Quality assurance ensures HSC number and viability as well as transduction efficiency prior to patients receiving full myeloablative conditioning. Gene-modified cells are transplanted into the conditioned recipient and then trafficked into the brain, where they engraft as microglial-like cells and thus deliver enzyme effectively to brain cells.

**Figure 2 ijms-23-04854-f002:**
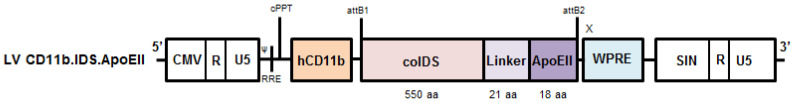
The CD11b.IDSApoEII lentiviral construct lentiviral vector under the CD11b promoter encoding for the codon-optimised human IDS gene followed by a flexible linker and the ApoEII peptide as a tandem repeat [57].

**Table 1 ijms-23-04854-t001:** Summary of the main conclusions and limitations of available literature assessing the neurological impact of HSCT.

Author and Year	Number of Patients and Age at HSCT	Conclusions	Issues
Vellodi et al., 1999 [32]	10 patients Age: 10 months–5 years 1 month	Possible role for BMT in the young asymptomatic child No benefit if neurological impairment at time of transplant More reliable predictors of phenotype urgently required	High TRM, variable age at HSCT, variable clinical phenotype, 1 of the 3 surviving patients transplanted with carrier donor
Maria et al., 2007 [33]	5 patients Age: 3 months– 3 years 4 month	Four out of five patients engrafted with full donor chimerism post umbilical cord blood HSCT All showed gains in cognitive, language, adaptive and motor skills, with the oldest patient having the slowest gains	Long term follow-up data needed
Guffon et al., 2009 [34]	8 patients Age: 3 years–16 years	BMT did not modify neurological deterioration in patients transplanted with severe phenotype Two patients transplanted with attenuated (non-neuropathic) form achieved adulthood with normal IQ, schooling and social development, and no language impairment Seven out of eight patients alive between 7 and 17 years post HSCT One death in cohort occurred over 6 years post HSCT and from unrelated cause	Patients with severe phenotype had significant cognitive impairment at time of HSCT, all patents aged over 3 years at time of transplant, 2 patients with severe form transplanted from heterozygous siblings
Poe et al., 2011 [35]	9 patients Age: 1.5 months–3 years 11 months	Improved neurological outcomes compared to untreated patients, although some developmental delays still apparent Five of the seven living patients (7 months to 7 years follow up) continuing to show gains in some or all of the developmental domains evaluated with one having normal development in 4 out of 6 domains	Unclear neurological status pre-HSCT and whether any correlation between age at time of transplant, expected phenotype and outcome
Escolar et al., 2012 [36]	9 patients Age: 1.5 to 47 months	Patients undergoing umbilical cord HSCT before 18 months of age showed continuous gains in cognitive, adaptive and language skills achieving very close to normal levels Boys transplanted >18 months old reached plateau before regressing to functional age between 1 and 3 years	Data only available up until patients 8 years old, more consistent data needed
Tanaka et al., 2012 [37]	21 patients Age: 2 years to 19 years 8 months	HSCT is effective for brain involvement if performed before the onset of developmental delay and cerebral atrophy although perhaps not for the most severe forms HSCT is associated with stabilisation and some improvement in cardiac valve dysfunction	Retrospective data, all patients aged over 2 years at HSCT
Annibali et al., 2012 [38]	4 patients Age: 2 years 6 months to 2 years 11 months	Improvement or stabilisation in somatic symptoms Neurological regression much slower than expected	Patients had mild to moderate mental retardation prior to HSCT
Wang et al., 2016 [39]	12 patients Age: 2–6 years	Patients transplanted between 2 and 6 years showed some improvement in motor and speech skills Overall outcomes of cardiac involvement, neurodevelopment and orthopaedic complications unclear	Short follow up (only 2 years), patients evaluated as part of larger MPS cohort so making conclusions applicable to MPSII challenging
Kubaski et al., 2017 [40]	27 patients Age: 2–21.4 years	HSCT good therapeutic option for MPS II and effective in resolving broad range of clinical outcomes Modest improvements in neurological outcomes for older patients treated with HSCT but better than those treated with ERT	Major limitation of study age at time of transplant, need more data on patients transplanted under 2 years old
Selvanathan et al., 2017 [41]	4 patients Age: 8 months–3 years, 8 months	All 4 patients showed neurocognitive stabilisation with 3 out of 4 showing improvement Most benefit at younger age of transplant and prior to neurological sequelae All patients have ongoing musculoskeletal problems	Varying pre-HSCT baselines make it difficult to draw any significant conclusions

**Table 2 ijms-23-04854-t002:** Current and potential problems with the therapeutic strategies for MPSII alongside potential solutions.

Current and Potential Problems with the Therapeutic Strategies for MPSII	Potential Solutions
Better outcomes associated with diagnosis and initiation of therapy at young age prior to onset of disease manifestations	Newborn screening
Improved outcomes correlate with higher enzyme levels	Supra-physiological enzyme levels achieved with HSCGT by using specific promoter (CD11b) to alter transcriptional control of the transgene
Obligate delay in time taken for engrafted HSCGT cells to differentiate into microglia and correct enzyme deficiency in the CNS	Use of ApoE peptide in HSCGT vector to increase ability of somatic enzyme to cross BBB
High morbidity and mortality associated with allogeneic stem cell transplant for MPSII	Autologous HSCGT safer and avoids risk of GVHD and need for immune suppression in conditioning protocols
High cost of delivering HSCGT	Durable, life-long clinical benefits expected from successful engraftment of a single infusion of genetically-modified HSCs and subsequent therapeutic gene expression by their progeny will mitigate the need for regular expensive ERT infusions
Lifestyle impact of frequent hospital attendances for ERT	HSCGT potentially abrogates the need for ERT if therapeutic enzyme levels are achieved

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
