# Peer review of "Current and Future Treatment of Mucopolysaccharidosis (MPS) Type II: Is Brain-Targeted Stem Cell Gene Therapy the Solution for This Devastating Disorder?"

_ijms, 2022, doi:10.3390/ijms23094854_

Round 1

Reviewer 1 Report

This author group provides an overview of the pathology of MPS II and current treatment strategies, which includes a detailed review of HSCT outcomes for patients with MPS II. The author group also reviews information on HSCT gene therapy and how it applies to this patient population. An update on MPS II is critically important in a rapidly evolving therapeutic landscape. Thank you for the opportunity to review with well-written manuscript that is well organized, updated, and complete. I have some notes for the team:

Line 56-58: The velocity of developmental/intellectual regression in severe MPS II patients has been shown to be more complicated to calculate due to the prolonged plateau of developmental stagnation that can last years. It might be worth noting how drawn out this period can be for these patients.

Line 59: When comparing severe MPS II patients to those with severe MPS I, the authors note that patients with MPS II can be “hyperactive and aggressive”. While this is certainly true, there has been some work done to parse out what contributes to these behavioral manifestations that provides some important clarification with regard to the behavioral phenotype. Specifically, it has been posited that the aggressive behavior observed in patients with severe MPS II may be caused by a combination of frustration, limited communication skills, anxiety, sensory-seeking behavior, and poor emotional regulation (Roberts, Steward, Kearney, 2016; Eisengart et al., 2020).]

Line 70-71: While learning difficulties can be common among patients with attenuated MPS II, there is also evidence for broader neurocognitive difficulties outside of the educational environment, including deficits in attention and visual-motor skills.

Line 128: This is the first place the term “neuronopathic” was used in the manuscript. For clarity and consistency, I recommend briefly describing this term in the Introduction section and noting when this term will be used (e.g., when reviewing articles that have used that term) versus when “severe” will be used.

Line 248: When describing the neurodevelopmental outcomes of the McKinnis et al (1996) report, I think it would be most helpful to report on the fact that at 8 years old, the patient’s intellectual level was consistent with that of a 10-month-old. Standardized scores for IQ measures at these very low levels do not provide much information, especially when patients are hitting the floor of the measure (i.e., obtain the lowest possible standard score).

Line 264: When reviewing the Vellodi et al (1999) paper, I’d recommend updating terminology to be more consistent with current use of terms and to avoid stigmatizing terms. Rather than “mental handicap” (even though this is what the authors used) I encourage the authors to use “intellectual disability”, as an example.

Lines 333-334: Similar to the suggestion above, I recommend avoiding outdated and/or stigmatizing terms and suggest use of “intellectual disability” in place of “mental retardation”.

Author Response

Response to reviewer 1 comments:

Thank you for your helpful and valid points that have improved the manuscript. Please see detailed response to your suggestions below.

Line 56-58: The velocity of developmental/intellectual regression in severe MPS II patients has been shown to be more complicated to calculate due to the prolonged plateau of developmental stagnation that can last years. It might be worth noting how drawn out this period can be for these patients.

Response: In severe cases, developmental delay is usually apparent by 18 to 24 months, with slow progress after this stage and a developmental plateau between the age of 3-5 years. The velocity of regression observed can be more complicated to predict due to the prolonged plateau of developmental stagnation that can last several years.

Line 59: When comparing severe MPS II patients to those with severe MPS I, the authors note that patients with MPS II can be “hyperactive and aggressive”. While this is certainly true, there has been some work done to parse out what contributes to these behavioral manifestations that provides some important clarification with regard to the behavioral phenotype. Specifically, it has been posited that the aggressive behavior observed in patients with severe MPS II may be caused by a combination of frustration, limited communication skills, anxiety, sensory-seeking behavior, and poor emotional regulation (Roberts, Steward, Kearney, 2016; Eisengart et al., 2020).]

Response: In contrast to the characteristically placid nature of children severely affected with MPSI, patients with MPSII can be hyperactive and aggressive. Experts have sought to ascertain more about the nature of this behavioural phenotype and have suggested that limited communication skills, frustration, anxiety, sleep disturbance, sensory-seeking behaviour and poor emotional regulation all contribute (1, 2)

Line 70-71: While learning difficulties can be common among patients with attenuated MPS II, there is also evidence for broader neurocognitive difficulties outside of the educational environment, including deficits in attention and visual-motor skills.

Response: They are still, however, subject to the same multisystem pathological processes that occur in severe forms and may still have symptoms and complications leading to significant morbidity and disability. These include mild to moderate learning difficulties and broader neurocognitive struggles outside of the educational environment, such as deficits in attention and visual-motor skills.

Line 128: This is the first place the term “neuronopathic” was used in the manuscript. For clarity and consistency, I recommend briefly describing this term in the Introduction section and noting when this term will be used (e.g., when reviewing articles that have used that term) versus when “severe” will be used.

Response: (Addition to end paragraph 6 of intro (line 73))….. The term ‘neuronopathic’ is used by some authors and investigators when referring to the severe phenotype of MPSII, typically associated with the characteristic neuropathology experienced by affected individuals. For purposes of this review, severe disease will be referred to as severe throughout apart from when referring to articles that have used the term ‘neuronopathic’ in which cases ‘neuronopathic’ will be used.

Line 248: When describing the neurodevelopmental outcomes of the McKinnis et al (1996) report, I think it would be most helpful to report on the fact that at 8 years old, the patient’s intellectual level was consistent with that of a 10-month-old. Standardized scores for IQ measures at these very low levels do not provide much information, especially when patients are hitting the floor of the measure (i.e., obtain the lowest possible standard score).

Response: Serial biopsies demonstrated persistent GAG deposition in neural structures in contrast to reduction in GAG deposition observed in non-neural structures, and the patient suffered from a progressive neurological decline such to the extent that at 8 years old the patient’s intellectual level was consistent with that of a 10-month-old.

Line 264: When reviewing the Vellodi et al (1999) paper, I’d recommend updating terminology to be more consistent with current use of terms and to avoid stigmatizing terms. Rather than “mental handicap” (even though this is what the authors used) I encourage the authors to use “intellectual disability”, as an example.

Response: Of the 3 patients who survived more than 7 years, 2 patients continued to show steady progressive physical and intellectual disabilities.

Lines 333-334: Similar to the suggestion above, I recommend avoiding outdated and/or stigmatizing terms and suggest use of “intellectual disability” in place of “mental retardation”.

Response: All patients had mild or moderate intellectual disability prior to HSCT (IQs ranging from 49 to 70) and demonstrated improvement or stabilisation in somatic function post.

  1. Roberts J, Stewart C, Kearney S. Management of the behavioural manifestations of Hunter syndrome. Br J Nurs. 2016;25(1):22, 4, 6-30.
  2. Eisengart JB, King KE, Shapiro EG, Whitley CB, Muenzer J. The nature and impact of neurobehavioral symptoms in neuronopathic Hunter syndrome. Mol Genet Metab Rep. 2020;22:100549.

Reviewer 2 Report

Dear Authors,

Thank you for giving me this opportunity to review your paper. This manuscript reviews about the updated and upcoming clinical management. The content is comprehensive and useful information about the previous papers.

I have the following comment and request.

Related to the neonatal screening, the optimal screening has been started as a research trial in partial countries. The cutoff level for early treatment including urine uronic acid and genotype might be difficult. If the updated information about this point would be published, please add this description.

Author Response

Thank you for reviewing our article and your helpful and valid point to improve the manuscript

Related to the neonatal screening, the optimal screening has been started as a research trial in partial countries. The cutoff level for early treatment including urine uronic acid and genotype might be difficult. If the updated information about this point would be published, please add this description.

Response: no updated published information at this point but I have added reference for the addition of MPSII to RUSP

For this reason, patients need to be diagnosed at a young age which can be very challenging because of the varied presentation and relies on a high index of suspicion or newborn screening, which was recently approved in the US (1).

  1. Society NM. Advisory Committee on Heritable Disorders in Newborns and Children Votes to Approve MPS II for Recommended Uniform Screening Panel. 2022.